# Synthesis of Monolayer MoSe_2_ with Controlled Nucleation via Reverse-Flow Chemical Vapor Deposition

**DOI:** 10.3390/nano10010075

**Published:** 2019-12-31

**Authors:** Siyuan Wang, Guang Wang, Xi Yang, Hang Yang, Mengjian Zhu, Sen Zhang, Gang Peng, Zheng Li

**Affiliations:** 1School of Material Science and Engineering, Xiangtan University, Xiangtan 411105, China; 2College of Liberal Arts and Sciences, National University of Defense Technology, Changsha 410073, China; 3College of Advanced Interdisciplinary Studies, National University of Defense Technology, Changsha 410073, China

**Keywords:** transition metal dichalcogenides (TMDCs), MoSe_2_, controlled growth, reverse-flow chemical vapor deposition (CVD)

## Abstract

Two-dimensional (2D) layered semiconductor materials, such as transition metal dichalcogenides (TMDCs), have attracted considerable interests because of their intriguing optical and electronic properties. Controlled growth of TMDC crystals with large grain size and atomically smooth surface is indeed desirable but remains challenging due to excessive nucleation. Here, we have synthesized high-quality monolayer, bilayer MoSe_2_ triangular crystals, and continuous thin films with controlled nucleation density via reverse-flow chemical vapor deposition (CVD). High crystallinity and good saturated absorption performance of MoSe_2_ have been systematically investigated and carefully demonstrated. Optimized nucleation and uniform morphology could be achieved via fine-tuning reverse-flow switching time, growth time and temperature, with corresponding growth kinetics proposed. Our work opens up a new approach for controllable synthesis of monolayer TMDC crystals with high yield and reliability, which promote surface/interface engineering of 2D semiconductors towards van der Waals heterostructure device applications.

## 1. Introduction

Two-dimensional (2D) layered transition metal dichalcogenides (TMDCs) (MX_2_: M = W, Mo. X = S, Se, Te) [1,2] have triggered significant attention for their atomically thin geometry and novel band structures, which have shown promising applications in the fields of electronics, optoelectronics, valleytronics, as well as energy conversion and storage [3,4,5,6]. Most semiconducting TMDCs host a bandgap from visible to near-infrared (NIR) range, exhibiting a transition from indirect bandgap to direct bandgap with thickness reduced from multilayer to monolayer due to quantum confined effect [7,8]. Among various TMDCs, MoSe_2_ has a tunable bandgap in the range of 1.1–1.5 eV by the thickness [9,10], making it one of the most desirable candidates in photovoltaic single-junction solar cells and photoelectrochemical cells [11]. MoSe_2_ not only shows great potential for the application in field-effect transistors (FETs) and electronic logic devices but also provides an ideal platform in optoelectronics with its narrower bandgap, larger spin-splitting energy and higher optical absorbance than MoS_2_ [12,13,14,15,16].

Chemical vapor deposition (CVD) is a low-cost, fast and scalable technique to synthesize various TMDCs [17,18,19]. Although CVD-grown MoSe_2_ crystals, films, ribbons, and van der Waals heterostructures [20,21,22,23] has been widely reported for versatile applications in transistors, photodetectors and sensors [24,25,26], it is still a great challenge to controllably synthesize high-quality monolayer MoSe_2_ with large grain size and good uniformity simultaneously. In a typical CVD growth process, the synthesis of atomically thin crystals is delicate and sensitive to growth parameters. The uncontrolled nucleation leads to inhomogeneous morphology and apparent roughness, which hinder the reliable formation of large-size crystals and highly uniform continuous films with a clean surface.

Recently, a reverse-flow strategy was reported in the sequential vapor deposition growth process, which can effectively prevent undesired thermal degradation and uncontrolled homogeneous nucleation, thus enabling the controllable synthesis of TMDC crystals with high quality, yield, and reliability. Zhang et al. employed the reverse-flow strategy in the growth process, and realized highly robust epitaxial growth of diverse 2D lateral heterostructures, multiheterostructures, and superlattices [27]. Zhang et al. achieved in epitaxial growth of MoS_2_, WS_2_, ternary and quaternary bilayer single crystals by reverse-flow CVD [28]. This method is also applicable for controllable synthesis of other TMDC monolayer flakes and films on diverse substrates, by decreasing high reaction temperature and simplifying complex processes, indicates that controlled nucleation, high yield and reliability could be achieved via fine-tuning the growth parameters. However, the reverse-flow switching time, growth time and temperature, as well as how these experimental parameters influence the morphology, nucleation density, and optical properties of CVD-grown monolayer MoSe_2_ have not been systematically investigated.

Here, we have controllably synthesized high-quality, large-size monolayer MoSe_2_ by introducing the reverse-flow strategy to a one-step CVD process, which could prevent the unintended supply of the chemical vapor source to eliminate uncontrolled nucleation, without the assistance of any catalyst or promoter. High crystallinity, uniform morphology and excellent optical response of MoSe_2_ have been systematically investigated and carefully identified. Reverse-flow switching time, growth time and temperature were demonstrated to play key roles in the nucleation process and corresponding growth kinetics. The large modulation depth and low saturation strength indicating that as-grown MoSe_2_ film has the desired nonlinearity and shows a great saturated absorption performance.

## 2. Experimental Methods

### 2.1. Synthesis of MoSe_2_

MoSe_2_ was grown in a home-modified two-temperature zone furnace CVD system with a 2-inch quartz tube (see Figure 1a). Both sides of the quartz tube are equipped with a gas inlet and outlet. The direction of gas flow can be switched by turning on gas valves 1 and 4 and turning off gas valves 2 and 3 or vice versa. The growth process is divided into two continual growth stages as shown in Figure 1b. A reverse-flow from zone 2 to zone 1 is applied during the temperature ramping and stabilization in stage I before growth, while a forward flow from zone 1 to zone 2 is applied for the epitaxial growth in stage II. MoO_3_ (15 mg, Aladdin, 99.95%) and selenium (Se) powders (300 mg, Aladdin, 99.95%) were used as precursors. MoO_3_ was placed in a corundum boat at the center of heating zone 2 with 300 nm SiO_2_/Si substrate faced down on top of the boat, while Se powders were placed in another corundum boat at the center of heating zone 1. The distance between the two boats was 20 cm. Firstly 300 sccm Argon (Ar) gas was introduced for 10 min before growth in order to flush the gas line and remove oxygen, then the furnace was heated to the desired growth temperature at a ramping rate of 25 °C per minute from room temperature (RT) with a reverse-flow of 45 sccm Ar. After that, a mixture of Ar/H_2_ (45 sccm/5 sccm) was used as the carrier gas with a forward-flow and the reaction process lasted for 10–15 min. The flow direction and rate did not change during the growth until the furnace cooled down to RT naturally. The diagram of the growth temperature program is shown in Figure 1c.

### 2.2. Characterization of MoSe_2_

The surface morphology of MoSe_2_ was systematically examined by employing an optical microscope (OM, Nikon Inc., Tokyo, Japan), atomic force microscope (AFM, Bruker Inc., Beerlika, MA, USA), and scanning electron microscope (SEM, Hitachi Inc., Tokyo, Japan). Raman spectra and photoluminescence (PL) measurements were performed with a 532 nm laser (Renishaw Inc., Wotton underedge, UK). Crystalline stoichiometry was analyzed using an X-ray photoemission spectroscopy (XPS, Thermo Fisher Scientific Inc., Waltham, MA, USA) and X-ray diffraction instrument (XRD, Bruker Inc.). High-resolution transmission electron microscopy (HRTEM, JEOL Inc., Tokyo, Japan) operated at 200 kV was carried out to demonstrate the ideal crystallinity and atomic structure of MoSe_2_. Open-aperture Z-scan measurements were performed using a homemade mode-locked Yb fiber laser operating at 1064 nm with a pulse width of 15 ps and a repetition rate of 40 MHz. 

## 3. Results and Discussions

The synthesis of atomically thin TMDC crystals is delicate and sensitive to growth conditions. In a typical CVD growth process with a single forward gas flow direction, the uncontrolled nucleation hinders the reliable formation of large-size crystals and irregular morphology and poor uniformity surface easily appeared (see Appendix A). A reverse-flow strategy from the substrate to the source could prevent the unintended supply of the chemical vapor source to eliminate uncontrolled nucleation. However, if the reverse-flow is introduced immediately after the end of the growth, the sudden flux change may cause crystal flakes self-decompose into fragments (see Appendix A). Therefore, exploring the appropriate time to switch the reverse-flow is crucial for ensuring the controllable growth of homogeneous monolayer MoSe_2_. If the reverse-flow is switched to forward-flow before reaching the designed temperature (~30 min), it leads to inhomogeneous morphology and apparent roughness on the substrate with a high nucleation density ~10,600 mm^−2^, as shown in Figure 2a. As shown in Figure 2b, if the reverse-flow is prematurely switched to forward-flow before reaching the desired temperature (~10 min), the source vapor will quickly reach the downstream substrate and react, forming a rough multilayer and monolayer flakes with jagged edges coexisted. If the reverse-flow is switched ~5 min before the setpoint, the multilayer MoSe_2_ flakes will further decrease with a nucleation density ~8000 mm^−2^ (see Figure 2c). The reverse-flow was introduced during the temperature ramping stage and the optimum switching time was precisely regulated to be ~1 min before reaching the desired growth temperature, and then forward-flow was applied until the end of the growth. By effectively suppressing excessive nucleation and self-decomposition, monolayer MoSe_2_ flakes with uniform morphology and atomically smooth surface have been successfully synthesized with a lower nucleation density ~5700 mm^−2^ (Figure 2d). The nucleation density can be controlled by fine-tuning the reverse-flow switching time before growth, with the statistical results shown in Figure 2i. As the gas flow rate increased, the monolayer flakes merged with each other to form a continuous film.

The growth time and temperature were also demonstrated to play key roles in the nucleation process and corresponding growth kinetics. Firstly, we changed growth time and fixed other parameters, when growing 5 min, small-size MoSe_2_ flakes (2–3 μm) were observed (Figure 2e) with a high nucleation density ~11700 mm^−2^. By increasing the growth time to 10 min, homogeneous monolayer MoSe_2_ flakes could be synthesized with a moderate nucleation density ∼5700 mm^−2^ (Figure 2d), among which the largest one is up to 110 µm. Bilayer MoSe_2_ crystals appeared when the growth time increasing to 15 min and the nucleation density was reduced to ~5000 mm^−2^ (Figure 2f). The statistical variation of nucleation density with growth time was shown in Appendix A. The precursors volatilize as the growth time increases, fewer precursors may lead to lower supersaturation, which therefore results in lower nucleation density. Then, we changed the growth temperature and fixed other parameters. At 720 °C, small-size MoSe_2_ flakes (~5 μm) occurred with a nucleation density of ~6900 mm^−2^ (Figure 2g). By increasing the temperature to 760 °C, homogeneous monolayer MoSe_2_ flakes could be synthesized with a moderate nucleation density ~5700 mm^−2^ (Figure 2d). While at 810 °C, large-size monolayer and small-size multilayer flakes coexisted with a reduced nucleation density of ~3700 mm^−2^ (Figure 2h). The statistical variation of nucleation density with temperature was shown in Figure 2j. In general, the longer growth time usually leads to a lower nucleation density, while lower growth temperature typically results in a higher nucleation density. It is consistent with the nucleation model of the vapor phase deposition, that the nucleation probability is proportional to the supersaturation and inversely proportional to the temperature [29].

The nucleation density is well controlled by optimizing the experimental parameters such as reverse-flow switching time, growth time and temperature. Figure 3a shows a typical continuous and uniform MoSe_2_ film synthesized on the SiO_2_/Si substrate with the reverse-flow method (growth temperature: ~760 °C, mixed vapor concentration: 80 sccm Ar/5 sccm H_2_, growth time: 10 min). Monolayer MoSe_2_ became dominant during the CVD growth with the introduction of H_2_ [30]. Similarly, triangular monolayer MoSe_2_ flakes up to 110 μm and bilayer MoSe_2_ flakes can be controllably grown under the same temperature (760 °C) and a mixture of Ar/H_2_ (45 sccm/5 sccm) for 10–15 min (Figure 3b and Appendix A), revealing thermodynamically stable edge termination and the threefold symmetry of the unit cell. Most bilayer MoSe_2_ flakes exhibit an AA stacking structure (the relative rotation angle of two vertically stacked triangles θ = 0°), while a few AB stacking bilayer (the relative rotation angle θ = 60°), hexagonal bilayer and trilayer flakes can also be observed, which may indicate that AA stacking order is more favorable under lower growth temperature. Appendix A shows SEM images of monolayer and bilayer MoSe_2_ triangular flakes with clear surface and boundary, confirming smoothness and high uniformity of the sample. The height of MoSe_2_ film, monolayer, and bilayer crystals were measured ~0.7 nm, ~0.65 nm, and ~0.8 nm respectively by AFM (Figure 3c,d and Appendix A), which is comparable to exfoliated samples [12] and our previous works [22,23]. 

The crystallinity and optical properties of monolayer and bilayer MoSe_2_ flakes were evaluated by employing Raman and PL measurements (Figure 3e–f). There are two characteristic peaks of Raman spectra of MoSe_2_, which correspond to the out-of-plane A_1g_ and in-plane $E_{2g}^{1}$ vibration mode. The A_1g_ peaks for monolayer and bilayer MoSe_2_ are located at 240 cm^−1^ and 241 cm^−1^, and the $E_{2g}^{1}$ peaks for monolayer and bilayer MoSe_2_ are located at 287 cm^−1^ and 285 cm^−1^, respectively. The A_1g_ peak is blue-shifted due to increasing interlayer coupling, and $E_{2g}^{1}$ peak is red-shifted may result from long-range coulomb interlayer interactions from monolayer to bilayer, the peak spacing between A_1g_ and $E_{2g}^{1}$ mode decreases as layer number increases from monolayer (47 cm^−1^) to bilayer (44 cm^−1^), which is consistent with previous results [9,10,20,31]. The monolayer MoSe_2_ occurs a prominent emission peak at ~807 nm (A-exciton) in the PL spectrum, which is much higher than the intensity of bilayer MoSe_2_ whose red-shifted peak at ~ 818 nm by more than 10 times (Figure 3f), indicating that the transition from a direct bandgap of monolayer MoSe_2_ at 1.54 eV to an indirect bandgap of bilayer MoSe_2_ at 1.51 eV, distinctly different from the bulk bandgap of ~1.1 eV [9,10]. Raman intensity maps ~240 cm^−1^ (A_1g_ mode) of the monolayer MoSe_2_ films and flakes reveal a particular uniform color contrast, verifying the highly uniform surface morphology and crystalline quality cover across the substrate (Figure 3g–h). The optical images, Raman (A_1g_ mode) and PL peak (807 nm) intensity maps of AA stacking bilayer MoSe_2_ have shown a strong contrast (Appendix A) and reveal the distinct boundary between the monolayer and bilayer regions. The optical images and Raman peak (A_1g_ and $E_{2g}^{1}$ mode) intensity maps of a hexagonal MoSe_2_ flake are also shown (Appendix A), reflecting that the second layer triangles begin to grow from the same nucleation site at the center of the first layer flakes.

We have further carried out XPS measurements to identify the stoichiometry and elemental composition of monolayer MoSe_2_. The binding energies of Mo 3d_3/2_ and Mo 3d_5/2_ are ~232.4 eV and ~229.3 eV (Figure 4a), and the binding energies of Se 3d_5/2_ and Se 3d_3/2_ are ~54.9 eV and ~55.8 eV (Figure 4b) respectively, in good agreement with previously reported values [10,20]. The atomic ratio of Mo and Se calculated from the XPS spectra is 1:1.92, which is very close to the stoichiometry of MoSe_2_. The crystalline stoichiometry was further confirmed by an XRD spectrum as shown in Figure 4c. The monolayer MoSe_2_ film was transferred to a TEM grid with the aid of polymethyl methacrylate (PMMA) solution. Figure 4d,e display bright-field TEM images of continuous MoSe_2_ film and triangle MoSe_2_ flakes, respectively. The high crystallinity of MoSe_2_ was further demonstrated by a high-resolution TEM (HRTEM) image and the fast Fourier transformation (FFT) pattern in the inset (Figure 4f). The FFT pattern clearly reveals as-transferred MoSe_2_ film is a single crystal with a hexagonal lattice structure. The lattice spacing measured from the atomically resolved TEM image is ~0.28 nm, corresponding to the (10$\bar{1}$0) plane [9,10,20].

The optical modulator plays an important role in the pulsed laser generation system. MoSe_2_ has been previously studied as broadband optical modulator materials for pulsed fiber laser systems due to their strong light-matter interaction [14−16]. The nonlinear optical absorption of CVD-grown MoSe_2_ and exfoliated samples is usually weakened by their random morphology, uncontrollable size and non-uniform thickness. Large-size monolayer MoSe_2_ flakes (Appendix A) and uniform continuous film (Appendix A) can also be grown on the sapphire substrate with good light transmittance [10,22,23,32,33]. Open-aperture Z-scan measurements were performed using a homemade mode-locked Yb fiber laser operating at 1064 nm with a pulse width of 15 ps and a tunable repetition rate of 40 MHz [34]. The experimental setup is shown in Figure 5a and we further investigated the nonlinear optical absorption properties of as-grown MoSe_2_ films (Figure 5b). The laser beam was split into two parts: one part was measured by a power meter, and the other was focused by a lens with a focal length of 75 mm onto the samples with the focused beam waist estimated to be ~80 μm and Rayleigh length of ~2.0 mm. The transmitted laser was focused by a lens with a focal length of 100 mm and measured by another power meter. The sample was fixed to a stepper motor, and its movement was controlled by a computer program. The nonlinear saturable absorption parameters can be obtained by fitting the Z-scan curves. The fitting formula of the open aperture Z-scan curves could be described as [35]:(1)$$T=\left(1-\frac{∆R\times I_{S}}{I_{S}+\frac{I_{0}}{1+\frac{z^{2}}{z_{0}^{2}}}}\right)/\left(1-∆R\right)$$
where *T* is normalized transmittance, and ∆*R*, *I*_S_, *I*_0_ and *Z*_0_ are the modulation depth, saturation fluence, incident pulse fluence and Rayleigh length of the incident laser beam, respectively. According to the fitting curve (Figure 5c), the saturation strength *I_S_* was fitted to be 110 MW/cm^2^, corresponding to the modulation depth ∆*R* of around 38.2%, indicating the desired nonlinearity. The large modulation depth and low saturation strength show a good saturated absorption performance, indicating that the MoSe_2_ film has great potential as an excellent saturable absorber in a pulsed laser generation system.

## 4. Conclusions

In conclusion, we have developed a general and reliable reverse-flow CVD strategy for controllable synthesis of high-quality monolayer MoSe_2_ crystals up to 110 μm, as well as uniform films. The excessive nucleation density was suppressed, while high crystallinity and uniform morphology of MoSe_2_ were carefully demonstrated. The optimal growth was achieved via precisely controlling reverse-flow switching time, growth time and temperature. The modulation depth and saturation strength of MoSe_2_ on sapphire substrates were measured to be 38.2% and 110 MW·cm^2^ respectively, the desired nonlinearity enables it as an ideal optical modulator material for pulsed laser and photodetectors. This work paves a new way for the controllable growth of 2D TMDCs and facilitates the development of electronic and optoelectronic applications.

## Figures and Tables

**Figure 1 nanomaterials-10-00075-f001:**
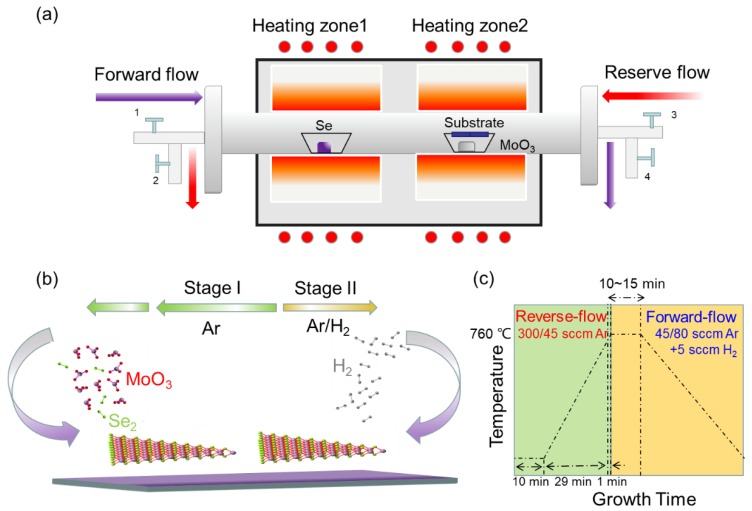
(**a**,**b**) Schematic illustration of two-temperature zone furnace reverse-flow chemical vapor deposition (CVD) setup and two growth stages. (**c**) The diagram of the growth time and temperature.

**Figure 2 nanomaterials-10-00075-f002:**
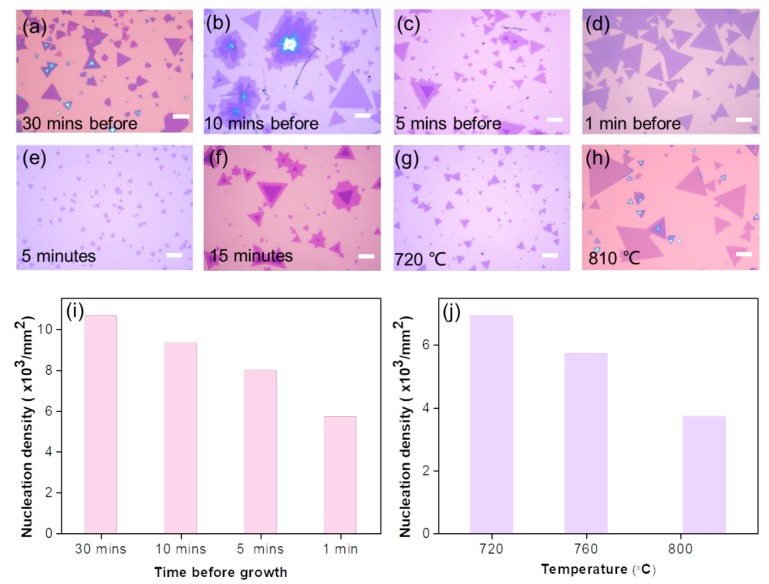
(**a**–**d**) Optical images of MoSe2 samples grown at 760 °C for 10 min, when the switch time turning reverse- to forward-flow (**a**) 30 min (**b**) 10 min (**c**) 5 min (**d**) 1 min before reaching 760 °C, respectively. (**e**–**h**) Optical images of MoSe2 samples when the switch time turning reverse- to forward-flow (Ar/H_2_, 45 sccm/5 sccm) 1 min before reaching growth temperature. The growth time is (**e**) 5 min and (**f**) 15 min at 760 °C, respectively. The growth temperature is (**g**) 720 °C and (**h**) 810 °C, with 10 min growth time. Statistical variation of nucleation density with (**i**) growth time and (**j**) temperature. A mixture of Ar/H_2_ (45 sccm/5 sccm) was applied in (**a**–**h**). The scale bar is 10 μm.

**Figure 3 nanomaterials-10-00075-f003:**
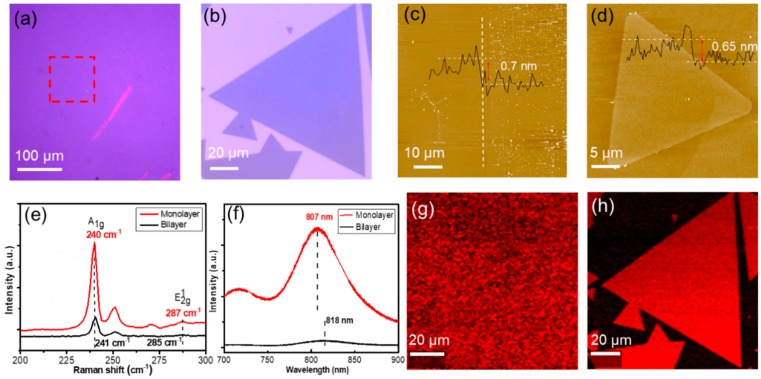
(**a**) The optical image of a monolayer MoSe_2_ film grown on SiO_2_/Si substrate. (**b**) The optical image of a monolayer MoSe_2_ flake. (**c**,**d**) atomic force microscope (AFM) topography of the monolayer MoSe_2_ film and flake. (**e**) Raman and (**f**) PL spectra of the monolayer and bilayer MoSe_2_, respectively. (**g**,**h**) Raman intensity maps ~ 240 cm^−1^ (A_1g_ mode) of the monolayer MoSe_2_ film and flake, respectively.

**Figure 4 nanomaterials-10-00075-f004:**
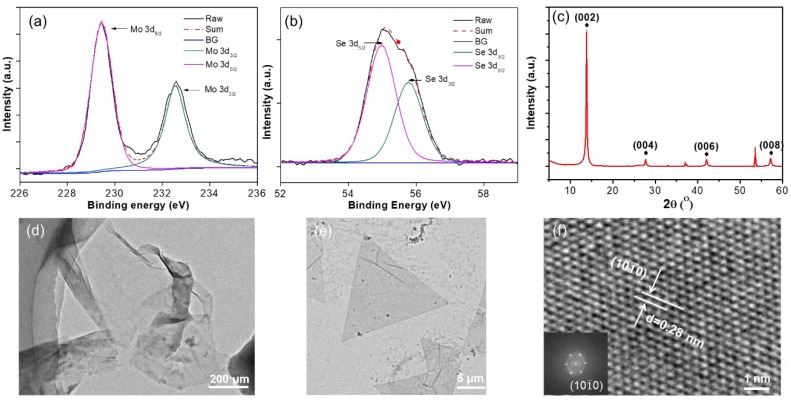
The X-ray photoemission spectroscopy (XPS) spectra of the monolayer MoSe_2_ film, where the (**a**) Mo 3d and (**b**) Se 3d binding energies are identified. (**c**) The X-ray diffraction instrument (XRD) spectrum of the monolayer MoSe_2_ film. (**d**) transmission electron microscopy (TEM) image of the transferred monolayer MoSe_2_ film and (**e**) MoSe_2_ flakes. (**f**) High-resolution transmission electron microscopy (HRTEM) image shows the lattice structure of the monolayer MoSe_2_ film with its corresponding fast Fourier transformation (FFT) pattern (inset).

**Figure 5 nanomaterials-10-00075-f005:**
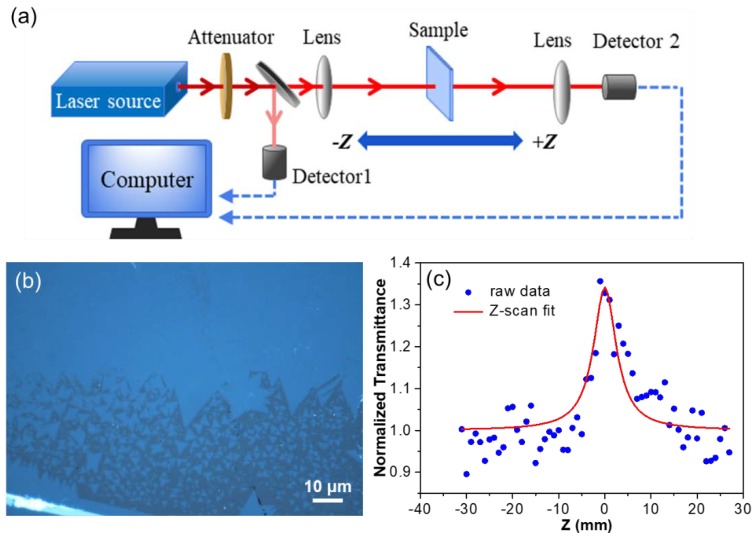
(**a**) The schematic diagram of the open-aperture Z-scan measurement system setup. (**b**) Optical image of monolayer MoSe_2_ films grown on sapphire (0001) substrate. (**c**) Normalized transmittance and Z-scan fitting results.

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
