# Peer review of "Synthesis of Monolayer MoSe2 with Controlled Nucleation via Reverse-Flow Chemical Vapor Deposition"

_nanomaterials, 2019, doi:10.3390/nano10010075_

Round 1

Reviewer 1 Report

The controlled growth of uniform 2D materials remains challenging due to excessive/undesired nucleations. Reverse-flow CVD has been proposed in recent studies to control undesired nucleations. However, a systematic study of the influence of the growth parameters in the proposed reverse-flow approach has not yet been carried out to date. This manuscript, therefore, investigates the influence of experimental parameters of the reverse-flow system such as switching time, growth time and temperature on the morphology, nucleation density and optical properties of monolayer MoSe2. The findings are very informative and appealing to the wide readership of Nanomaterials. I, therefore, recommended the manuscript for publication in Nanomaterials after a minor revision.  

Review Comment (s)

Reverse is wrongly spelled in the title (revere-flow) of the manuscript; the authors should rectify this typo accordingly.

The authors hypothesized on page 4 lines 141-145 that, “In general, the longer growth time usually leads to a lower nucleation density, while lower growth temperature typically results in a higher nucleation density. It is consistent with the nucleation model of the vapor phase deposition, that the nucleation probability is proportional to the supersaturation and inversely proportional to the temperature (Reference 31, Nanotechnology, 2018, 29: 314001 )”.

However, the cited paper made no mention of the time and temperature-dependent nucleation density of 2D materials. The authors should check and cite the appropriate paper.

Please explain why there is an increase in nucleation density with short growth time as observed in Figure 2e?

In Supporting Figure S4b, the authors claim the image in S4b is MoSe2 film, but it clearly appears as high-density MoSe2 To avoid any ambiguity, the authors should replace the image with a representative image of MoSe2 film on Sapphire (0001).

Author Response

Response to Reviewer 1 Comments

The controlled growth of uniform 2D materials remains challenging due to excessive/undesired nucleations. Reverse-flow CVD has been proposed in recent studies to control undesired nucleations. However, a systematic study of the influence of the growth parameters in the proposed reverse-flow approach has not yet been carried out to date. This manuscript, therefore, investigates the influence of experimental parameters of the reverse-flow system such as switching time, growth time and temperature on the morphology, nucleation density and optical properties of monolayer MoSe2. The findings are very informative and appealing to the wide readership of Nanomaterials. I, therefore, recommended the manuscript for publication in Nanomaterials after a minor revision.

Point 1: Reverse is wrongly spelled in the title (revere-flow) of the manuscript; the authors should rectify this typo accordingly.

Response 1: Thank you very much for the careful review and kind reminder. We sincerely apologize for the spelling mistake. The “revere-flow” in the title has been revised to “reverse-flow”. We have also examined and rectified this kind of typos accordingly in the manuscript.

Point 2: The authors hypothesized on page 4 lines 141-145 that, “In general, the longer growth time usually leads to a lower nucleation density, while lower growth temperature typically results in a higher nucleation density. It is consistent with the nucleation model of the vapor phase deposition, that the nucleation probability is proportional to the supersaturation and inversely proportional to the temperature (Reference 31, Nanotechnology, 2018, 29: 314001)”.

However, the cited paper made no mention of the time and temperature dependent nucleation density of 2D materials. The authors should check and cite the appropriate paper.

Response 2: Thank you very much for the careful review and helpful advice. We are sorry about the incorrect citations. The “reference [31]” in line 149 of page 4 should be “reference [29]”, the “reference [30]” in line 183 of page 6 should be “reference [31]”, and the references [30] and [31] are interchanged in line 315-318 of page 9. We have also revised these citation errors accordingly in the manuscript.

Point 3: Please explain why there is an increase in nucleation density with short growth time as observed in Figure 2e?

Response 3: Thank you very much for the constructive comments. We have noticed that there is a statement “All of our observations are consistent with the nucleation model of the vapor phase deposition developed by W. K. Burton and N. Cabrera, where they predict that the nucleation probability is proportional to the supersaturation and inversely proportional to the substrate temperature.” in a reference. (Zhou H, Wang C, Shaw J C et al., Large area growth and electrical properties of p-type WSe2 atomic layers. Nano Letters 15: 709 (2014)).

Compared with our results, we conclude that the saturation is proportional to the nucleation density, therefore we interpret it as “The precursors volatilize as the growth time increases, fewer precursors may lead to lower saturation, while the saturation is proportional to the nucleation density, which therefore results in lower nucleation density.” Hope it makes sense.

Point 4: In Supporting Figure S4b, the authors claim the image in S4b is MoSe2 film, but it clearly appears as high-density MoSe2 flakes. To avoid any ambiguity, the authors should replace the image with a representative image of MoSe2 film on Sapphire (0001).

Response 4: Thank you very much for the careful review and valuable suggestions. To avoid possible ambiguity of the image shown in Figure S4b which appears as dense MoSe2 flakes, we have replaced it with a representative image of MoSe2 film on Sapphire (0001).

Reviewer 2 Report

In the manuscript "Synthesis of monolayer MoSe2 with controlled nucleation via reverse-flow chemical vapor deposition" the authors present a comprehensive study of the parameters necessary to grow MoSe2 through reverse flow CVD. This work can have a strong impact for the readership interested in 2D materials. In particular, more and more research groups are developing CVD growth systems of TMDCs and therefore providing a thorough comparison between the different growing parameters and the  morphology of the growth samples can be cruzial for many readers.

Nonetheless in order to be even more interesting for the readership I really believe that the authors should do an effort to compare the outcome with their growth method with respect to other CVD results (even conventional results, not using reverse flow) to grow MoSe2. A critical comparison pointing out the  strengths and the weaknesses of the different methods might be necessary to convince the readership to embrace the presented growth method.

I also fund a bit weak the structural characterization of the as grown material. Only a small area TEM (and its FFT) is shown. Larger area TEM would be necessary in order to clearly state that the grown triangles are single-crystal. XRD analysis would also be very interesting to make this point stronger.

In view of this I suggest to revise the manuscript before its publication in order to put the results obtained here in context.

Author Response

Response to Reviewer 2 Comments

In the manuscript “Synthesis of monolayer MoSe2 with controlled nucleation via reverse-flow chemical vapor deposition” the authors present a comprehensive study of the parameters necessary to grow MoSe2 through reverse flow CVD. This work can have a strong impact for the readership interested in 2D materials. In particular, more and more research groups are developing CVD growth systems of TMDCs and therefore providing a thorough comparison between the different growing parameters and the morphology of the growth samples can be cruzial for many readers.

Point 1: Nonetheless in order to be even more interesting for the readership I really believe that the authors should do an effort to compare the outcome with their growth method with respect to other CVD results (even conventional results, not using reverse flow) to grow MoSe2. A critical comparison pointing out the strengths and the weaknesses of the different methods might be necessary to convince the readership to embrace the presented growth method.

Response 1: Thank you very much for the constructive comments and valuable suggestions. In a typical CVD growth process with a single forward gas flow direction, the uncontrolled nucleation hinders the reliable formation of large-size crystals and irregular morphology and poor uniformity surface easily appeared (see Figure S1a, S1b). A reverse-flow strategy from the substrate to the source could prevent the unintended supply of the chemical vapor source to eliminate uncontrolled nucleation. However, if the reverse-flow is introduced immediately after the end of the growth, the sudden flux change may cause crystal flakes self-decompose into fragments (see Figure S1c). Therefore, exploring the appropriate time to switch the reverse-flow is crucial for ensuring the controllable growth of homogeneous monolayer MoSe2. If the reverse-flow is switched to forward-flow before reaching the designed temperature (~ 30 minutes), it leads to inhomogeneous morphology and apparent roughness on the substrate with a high nucleation density ∼10600 mm-2, as shown in Figure 2a. The corresponding revision has been made from line 108 to 117 of page 3.

Point 2: I also fund a bit weak the structural characterization of the as grown material. Only a small area TEM (and its FFT) is shown. Larger area TEM would be necessary in order to clearly state that the grown triangles are single-crystal. XRD analysis would also be very interesting to make this point stronger.

Response 2: Thank you very much for the constructive comments and valuable suggestions. TEM image of the transferred monolayer triangle MoSe2 flakes has been added in Figure 4e, which can illustrate the grown triangles are single-crystal. In addition, XRD spectrum of the as-grown MoSe2 film has been also added in Figure 4c, which further confirms the crystalline stoichiometry.

Round 2

Reviewer 2 Report

In the revised version of the manuscript the authors have not fully address the main point raised in my referee report. In particular the authors have compared their results with those obtained with conventional CVD in their lab. But the readership would expect also a comparison with the results published in the literature. Quickly  searching on the internet I can find very large single crystals of MoSe2 grown with what looks conventional CVD. The authors should compare the results obtained in their lab with those reported by other groups.

Once this is incorporated to the manuscript I am confident that it will become publishable.

Author Response

Response to Reviewer 2 Comments Point 1: In the revised version of the manuscript the authors have not fully address the main point raised in my referee report. In particular the authors have compared their results with those obtained with conventional CVD in their lab. But the readership would expect also a comparison with the results published in the literature. Quickly searching on the internet I can find very large single crystals of MoSe2 grown with what looks conventional CVD. The authors should compare the results obtained in their lab with those reported by other groups. Once this is incorporated to the manuscript I am confident that it will become publishable. Response 1: Thank you very much for the constructive comments and valuable suggestions. We are aware that large single crystals and monolayer film of MoSe2 can be grown by conventional CVD, however, some excessive nucleation points may still appear surround the flake and on the surface, as shown in Figure R1(a) and R2(a), respectively (see reference [20] ACS Nano 2014, 8: 5125 and reference [21] RSC Advances 2017, 7: 27969). The uncontrolled nucleation easily leads to inhomogeneous morphology and apparent roughness, which hinder the reliable formation of large-size crystals and highly uniform continuous films with a clean surface. Therefore, it is still a great challenge to controllably synthesize high-quality monolayer MoSe2 with large grain size and good uniformity simultaneously. Here, we have controllably synthesized high-quality, large-size monolayer MoSe2 by introducing the reverse-flow strategy to a one-step CVD process, which could prevent the unintended supply of the chemical vapor source to eliminate uncontrolled nucleation, without the assistance of any catalyst or promoter. Compare with those results reported by other groups, our as-grown MoSe2 flake and film have optimized nucleation and uniform morphology, which are shown in Figure R1(b) and R2(b) respectively. The synthesis of atomically thin crystals is delicate and sensitive to growth parameters. High yield and reliability of monolayer MoSe2 synthesis with controlled nucleation could also be achieved via fine-tuning reverse-flow switching time, growth time and temperature. Furthermore, we demonstrate an optimized strategy to grow bilayer MoSe2 with controlled AA and AB stacking orders. Reverse gas flow effectively suppresses the random nucleation centers, leading to uniform growth of the second layer of MoSe2 on the first monolayer. Customized temperature profile selectively actives the growth of different MoSe2 bilayer. However, this is not the main research content of the current manuscript. The corresponding modification has been made in lines 46, 49 and 59 of page 2 and the reference [21] has been replaced in line 298-299 of page 9. Please see the attachment.
